# Cytological Comparison between Hepatocellular Carcinoma and Intrahepatic Cholangiocarcinoma by Image Analysis Software Using Touch Smear Samples of Surgically Resected Specimens

**DOI:** 10.3390/cancers14092301

**Published:** 2022-05-05

**Authors:** Sho Kitamura, Keita Kai, Mitsuo Nakamura, Tomokazu Tanaka, Takao Ide, Hirokazu Noshiro, Eisaburo Sueoka, Shinich Aishima

**Affiliations:** 1Department of Pathology, Saga University Hospital, Saga 849-8501, Japan; sn5388@cc.saga-u.ac.jp (S.K.); nakamum1@cc.saga-u.ac.jp (M.N.); saish@cc.saga-u.ac.jp (S.A.); 2Department of Surgery, Saga University Faculty of Medicine, Saga 849-8501, Japan; f8642@cc.saga-u.ac.jp (T.T.); idetaka@cc.saga-u.ac.jp (T.I.); noshiro@cc.saga-u.ac.jp (H.N.); 3Department of Clinical Laboratory Medicine, Saga University Faculty of Medicine, Saga 849-8501, Japan; sueokae@cc.saga-u.ac.jp; 4Department of Pathology and Microbiology, Saga University Faculty of Medicine, Saga 849-8501, Japan

**Keywords:** hepatocellular carcinoma, cholangiocarcinoma, cytology, touch smear, nuclear atypia

## Abstract

**Simple Summary:**

This study cytologically compared hepatocellular carcinoma (HCC) and intrahepatic cholangiocarcinoma (ICC) using image analyzing software. The results indicated that the major/minor axis ratio of ICC was significantly larger than in HCC in Papanicolaou staining. This difference was consistently observed in clinical samples of cytology such as fine-needle aspiration, brushing and ascites. This study indicated a significant difference in the nuclear morphology of HCC (round shape) and ICC (oval shape) in Papanicolaou-stained cytology specimens. This simple and objective finding is considered to be useful for differential cytodiagnosis of HCC and ICC.

**Abstract:**

To investigate useful cytological features for differential diagnosis of hepatocellular carcinoma (HCC) and intrahepatic cholangiocarcinoma (ICC), this study cytologically compared HCC to ICC using image analysis software. Touch smear specimens of surgically resected specimens were obtained from a total of 61 nodules of HCC and 16 of ICC. The results indicated that the major/minor axis ratio of ICC is significantly larger than that of HCC (1.67 ± 0.27 vs. 1.32 ± 0.11, *p* < 0.0001) in Papanicolaou staining. This result means that the nucleus of HCC is close to round and the nucleus of ICC is close to an oval. This significant difference in the major/minor axis ratio between ICC and HCC was consistently observed by the same analyses using clinical samples of cytology (4 cases of HCC and 13 cases of ICC) such a fine-needle aspiration, brushing and ascites (ICC: 1.45 ± 0.13 vs. HCC: 1.18 ± 0.056, *p* = 0.004). We also confirmed that nuclear position center-positioned nucleus (*p* < 0.0001) and granular cytoplasm (*p* < 0.0001) are typical features of HCC tumor cells compared to ICC tumor cells. The research study found a significant difference in the nuclear morphology of HCC (round shape) and ICC (oval shape) in Papanicolaou-stained cytology specimens. This simple and objective finding will be very useful for the differential cytodiagnosis of HCC and ICC.

## 1. Introduction

In Japan, 94.0% of primary liver cancers are hepatocellular carcinoma (HCC), 4.4% intrahepatic cholangiocarcinoma (ICC), and the remaining small proportion includes combined hepatocellular and cholangiocarcinoma or other rare tumors [1]. Although ICC is generally considered to be a rare tumor, it has been found that the prevalence of ICC is relatively high in limited regions worldwide including Japan and India [2]. It has also been reported that the development of ICC is related to hepatitis B and C virus infections, similar to that of HCC [3,4].

The usefulness of cytology for the investigation of hepatic nodules involving hepatocellular carcinoma (HCC) and intrahepatic cholangiocarcinoma (ICC) is limited at present. The development of imaging modalities, such as multiphasic computed tomography (CT), magnetic resonance imaging (MRI), and positron emission tomography (PET)-CT permits a noninvasive clinical diagnosis of classical HCC or ICC [5,6,7]. The cases with typical imaging would not require an invasive procedure, such as liver biopsy or fine-needle aspiration cytology (FNAC).

Liver biopsy or FNAC are generally considered useful diagnosis techniques for hepatic nodules without specific imaging features or for difficult cases of imaging diagnosis, such as small nodules (<2 cm) or indistinct nodules in cirrhotic patients [8]. Even in such cases, biopsy specimens are considered to be superior to FNAC samples from the viewpoint of tumor volume assessment and availability for immunohistochemistry. Therefore, FNAC for liver nodule investigation is rarely performed. However, an opportunity to distinguish between HCC and ICC on cytological material is encountered in greatly advanced/unresectable cases, in cases where the patient’s general condition is too poor to undergo a liver biopsy, and in cases with ascites or pleural fluid. Considering that the therapeutic strategy is different for HCC and ICC, it is important to distinguish HCC from ICC even with cytology alone.

The cytology of ICC is relatively familiar to cytopathologists compared to HCC because bile juice and brushings, which are obtained by endoscopic retrograde cholangiopancreatography or the biliary drainage route, are frequently submitted for cytological diagnosis [9,10]. Although the cytology of HCC is clinically rare, many studies have reported that the cytology of HCC is mostly obtained by FNAC [11,12,13,14,15,16,17,18,19,20,21,22]. These FNAC studies focused on cytologic differences between HCC and non-neoplastic hepatocytes or metastatic lesions. However, a definitive cytologic difference between HCC and ICC has not been well defined by researchers and therefore little knowledge has been accumulated regarding the differential diagnosis of HCC and ICC.

The aim of the present study was to investigate the cytological features that can facilitate the differential diagnosis of HCC and ICC, using touch smear samples of resected specimens and morphological analyses using image-analysis software.

## 2. Materials and Methods

### 2.1. Touch Smear Cytology

Seventy-seven hepatic nodules of 71 patients who underwent surgical resection in Saga University Hospital between 2010 to 2012 and 2020 to 2021 after a clinical diagnosis of HCC or ICC were enrolled in the study. Six patients underwent hepatic resection for 2 HCC nodules during 1 operation. The touch smears of hepatic nodules were obtained from fresh cut surfaces of resected specimens and then subjected to Giemsa and Papanicolaou staining. The details of the hepatic nodules were: HCC, 61 nodules (well differentiated; 8, moderately differentiated; 48, poorly differentiated; 5) and 16 ICC nodules (Table 1). The differentiation of HCC depended on pathological reports. Giemsa-stained touch smears were unavailable for 20 nodules. As a control sample, the touch smears (Papanicolaou staining only) of non-tumorous background liver were obtained from 5 cases. Comprehensive informed consent for the use of resected tissue for research was obtained from all patients, and the study protocol was approved by the Ethics Committee of Saga University Hospital (No. 2018-12-R-11).

### 2.2. Analysis of Hematoxylin and Eosin-Stained Tissue and Giemsa-Stained Touch Smear Cytology Using Imaging Software

To compare cytological findings, hematoxylin-eosin (HE) stained formalin-fixed paraffin-embedded resected specimens sliced into 4 μm sections were also analyzed in the present study. Three digital images of tumor tissue (×200) of HCC and ICC were analyzed using the imaging analysis software Tissue Studio (Definiens, München, Germany) and data on the major axis, minor axis and the area of the nucleus were automatically calculated (Figure 1a–d). In the same way, three digital images of Giemsa-stained touch smear cytology (×200) of HCC and ICC were also analyzed using Tissue Studio (Figure 2a,b). Each mean value of the major axis, minor axis, major/minor axis ratio and the area of the nuclei were statistically compared between HCC and ICC specimens.

### 2.3. Analysis of Papanicolaou-Stained Touch Smear Cytology Specimens Using Imaging Software

As the software Tissue Studio did not support Papanicolaou staining, analyses of Papanicolaou-stained touch smear cytology were performed using attaching software of EXpath III (INTEC, Toyama, Japan). Three digital images of Papanicolaou-stained touch smear cytology specimens (×200) of HCC and ICC were subjected to the analyses. In addition, Papanicolaou staining of touch smear cytology of non-tumorous hepatocytes was also analyzed. Twenty nuclei in each image were manually selected and then the major axis, minor axis and area of the nucleus were calculated (Figure 2c,d). Each mean value of the major axis, minor axis, major/minor axis ratio and the area of the nucleus were statistically compared for HCC, ICC, and non-tumorous hepatocytes.

### 2.4. Assessment of Cytological Findings of HCC and ICC in Papanicolaou-Stained Touch Smear Cytology Specimens

The cytological findings of HCC and ICC in Papanicolaou-stained touch smear cytology specimens were assessed by two authors (SK and KK) after discussion following observations under multi-headed microscope images. The following cytological findings were evaluated and categorized into several two titer classifications: nuclear contours (irregular vs. smooth), chromatin pattern (coarse/granular vs. fine), chromatin distribution (homogeneous vs. heterogeneous), nuclear position (center vs. uncentre), number of the nucleolus (single/unclear vs. multiple), cytoplasm (vacuole/foamy vs. granular) and cell boundaries (clear vs. unclear). Each finding was statistically compared between HCC and ICC.

### 2.5. Clinical Materials of Cytology for Validation

As the condition of clinical samples (such as bile juice, brushings, ascites, and fine-needle aspiration may be different from touch smear cytology, we performed the analysis of clinical materials of cytology for validation of results obtained by touch smear cytology of resected samples. A total of 17 samples in which tumor cells of HCC or ICC appeared were found (HCC: four cases, ICC: 13 cases) among 56,383 cytological samples examined at Saga University Hospital between 2014 and 2021. Each mean value of the major axis, minor axis, major/minor axis ratio and the area of the nuclei of tumor cells in Papanicolaou-stained specimens were statistically compared between HCC and ICC.

### 2.6. Statistical Analysis

The statistical analysis was performed using JMP (ver. 15.2 software, AS Institute, Cary, NC, USA). The comparisons of pairs of groups were performed using the Wilcoxon test or Fisher’s exact test (two-sided). Values of *p* < 0.05 were considered to be statistically significant findings.

## 3. Results

### 3.1. Comparison of Nuclei among HCC and ICC in HE-Stained Tissue Specimens

The results of comparisons of nuclei between HCC and ICC in each stained section are summarized in Table 2. In the HE-stained tissue specimens, the means and standard deviation (SD) of nuclei in HCC and ICC were evaluated thus: Major axis: HCC: 11.52 ± 2.98 μm vs. ICC: 14.12 ± 2.05 μm (*p* = 0.0003), Minor axis: HCC: 8.64 ± 1.91 μm vs. ICC: 9.45 ± 1.23 μm (*p* = 0.031), major/minor axis ratio: HCC: 1.36 ± 0.092 vs. ICC: 1.54 ± 0.083 (*p* < 0.0001), Nucleus area: HCC: 77.42 ± 38.44 μm^2^ vs. 93.64 ± 21.89 μm^2^ (*p* = 0.0099). The nuclei of ICC were significantly larger and oval-shaped rather than exhibiting a round shape like HCC found in HE-stained tissue specimens.

### 3.2. Comparison of Nuclei among HCC and ICC in Giemsa-Stained Touch Smear Cytology Specimens

In Giemsa-staining, means and SD of major axis, means of minor axis, major axis/minor axis ratio and area of nuclei were: major axis: HCC, 16.42 ± 4.47 μm vs. ICC; 18.46 ± 3.90 μm (*p* = 0.26), minor axis: HCC, 12.04 ± 2.93 μm vs. ICC 13.10 ± 2.26 μm (*p* = 0.29); major/minor axis ratio: HCC: 1.39 ± 0.10 vs. ICC; 1.44 ± 0.12 (*p* = 0.38); Nucleus area: HCC, 146.78 ± 68.40 μm^2^ vs. 176.32 ± 58.20 μm^2^ (*p* = 0.14). No significant difference between HCC and ICC was observed for all variables examined.

### 3.3. Comparison of HCC, CCC and Non-Tumorous Hepatocytes in Papanicolaou-Stained Touch Smear Cytology Specimens

In a comparison between HCC and ICC by Papanicolaou-stained touch smear cytology, means and SD of major axis, minor axis, major axis/minor axis ratio, and area of nuclei of HCC and ICC were: major axis, HCC, 6.89 ± 2.47 μm vs. ICC: 8.60 ± 3.06 μm (*p* = 0.057); Minor axis: HCC, 5.22 ± 1.79 μm vs. ICC, 5.22 ± 1.95 μm (*p* = 0.94); major/minor axis ratio: HCC, 1.32 ± 0.11 vs. ICC, 1.67 ± 0.27 (*p* < 0.0001), Nucleus area: HCC, 43.80 ± 27.31 μm^2^ vs. 55.37 ± 35.34 μm^2^ (*p* = 0.35). The nuclei of the ICC were significantly oval rather than round in shape compared to HCC cells, while no significant difference was observed in the nucleus area of Papanicolaou-stained touch smear cytology specimens.

Non-tumorous hepatocytes were also assessed, and the results are as follows: Major axis: 5.78 ± 0.97 μm; Minor axis, 4.77 ± 0.75 μm; major/minor axis ratio: 1.21 ± 0.11; Nucleus area, 36.08 ± 7.37 μm^2^. In comparison with HCC and ICC, the minor axis in non-tumorous hepatocytes appeared to be the smallest although statistical significance was not reached (vs. HCC: *p* = 0.56, vs. ICC: *p* = 0.71). A significant difference was found for the major/minor axis ratio in comparison to ICC (vs. HCC: *p* = 0.037, vs. ICC: *p* = 0.0015) (Figure 3). This result indicates the major/minor axis ratio of the nucleus (namely the shape of tumor nuclei is close to round or oval shapes) is an important finding for cytological differential diagnosis between HCC and ICC.

### 3.4. Comparison of Cytological Findings in Papanicolaou-Stained Touch Smear Cytology Specimens of ICC and HCC

Cytological findings of ICC and HCC in Papanicolaou-stained touch smear cytology specimens are summarized in Table 3. The nuclei of HCC were significantly center-positioned (*p* < 0.0001), having a single nucleolus rather than multiple nucleoli (*p =* 0.005), and a granular cytoplasm (*p* < 0.0001). No significant difference was observed in the chromatin pattern and chromatin distribution between HCC and ICC. Typical cytological images of HCC and ICC in Papanicolaou-stained touch smear cytology specimens are shown in Figure 4.

### 3.5. Comparison of HCC and CCC in Papanicolaou-Stained Clinical Specimens

We consider the findings regarding the major/minor axis ratio in Papanicolaou-stained touch smear cytology specimens to be very important as they may be useful in daily cytological practice. As the condition of clinical samples may be different from touch smear cytology specimens, we planned a validation study using Papanicolaou-stained clinical samples. The details of clinical samples are summarized in Table 4. From 2014 to 2021, we found only 4 cytological samples which contained HCC tumor cells. All 4 samples were obtained by FNA (liver: 3, lymph node:1). In addition, 13 clinical samples which contained ICC tumor cells were found during this time period (FNA: 3, brushings: 8, ascites: 2).

The means ± SD of each major axis, minor axis and major axis/minor axis ratio, and the area of nuclei of HCC and ICC were as follows: major axis: HCC, 8.79 ± 1.58 μm vs. ICC, 10.01 ± 1.90 μm (*p* = 0.308); Minor axis: HCC: 7.67 ± 1.32 μm vs. ICC, 7.00 ± 1.72 μm (*p* = 0.428), major/minor axis ratio: HCC, 1.18 ± 0.056 vs. ICC: 1.45 ± 0.13 (*p* = 0.004, Figure 5); Nucleus area: HCC, 72.00 ± 24.87 μm^2^ vs. 75.49 ± 27.47 μm^2^ (*p* = 0.821). These data are summarized in Table 5. Thus, the major/minor axis ratio was significantly different between HCC and ICC, even in clinical samples.

## 4. Discussion

The most notable finding of the present study was that the major/minor axis ratio of HCC and ICC was significantly different in Papanicolaou-stained cytological specimens. This means that the nuclei of HCC were close to round shapes and the nuclei of ICC close to oval shapes. This finding may have been previously noticed by other cytopathologists, but it is not well-recognized, presumably because no previous literature documenting this finding with analysis evidence has been published. This result was confirmed not only by the touch smear cytology of resected specimens but also by the validation of clinical cytological materials. Although our study also found a significant difference in the nuclear position, number of nucleoli and the cytoplasm between ICC and HCC, these cytological findings are subjective and depend on the experience of the investigating cytologist. Therefore, we consider that the major/minor axis ratio is very useful for the differential cytodiagnosis of ICC and HCC, because of its objective nature and simpleness. The significant difference in the major/minor axis ratio of ICC and HCC samples was confirmed in HE-stained tissue specimens, but its significance disappeared in Giemsa-stained specimens, likely because of morphological changes due to the dry process involved. The cell morphology is affected by the dry process of Giemsa-staining. Usually, the cell size is significantly enlarged, and cell morphology becomes more circular than Papanicolaou-staining. Therefore, it seems reasonable that the results of morphological analyses by image analyzing software were different between Giemsa-staining and Papanicolaou-staining. The significance of Papanicolaou-staining is important because clinical samples of HCC or ICC are usually evaluated by Papanicolaou-staining. Differential diagnosis of HCC and ICC is sometimes clinically problematic. The definitive diagnosis of HCC or ICC is usually made by pathological evaluation of morphology and immunohistochemical analyses of liver biopsy or resected specimens. Although it is rare, there are situations that require the differential diagnosis of HCC and ICC to be made only from cytological materials, when biopsy or surgical resection specimens could not be safely obtained because of the patient’s general physical condition (such as a bleeding tendency or cachexia). In this situation, knowledge of the major/minor axis ratio will greatly help the cytological differential diagnosis of HCC and ICC.

The cytological characteristics of HCC have been well documented by studies of FNAC [11,12,13,14,15,16,17,18,19,20,21,22]. HCC cells typically have granular cytoplasm but cytological findings such as cellularity, cell dissociation, cell borders, monotony, trabeculae structure, nucleoli, cell size, nucleus/cytoplasmic ratio, nuclear crowding, chromatin distribution, and nuclear pleomorphism vary widely according to the differentiation of HCC [13,16,17]. Regarding ICC, knowledge of cytological characteristics has been accumulated by FNAC and brushing cytology. Typical ICC cells cytologically show adenocarcinoma-like features such as three-dimensional clusters, foamy and/or vacuolated cytoplasm, loss of nuclear polarity and prominent nucleoli, although these features varied according to the stage of tumor differentiation [23,24,25,26].

Cytological differences between HCC and ICC have not been well documented by researchers and therefore a paucity of knowledge has been accumulated. We found only one article which tried to distinguish ICC and HCC using cytological findings. Sampatanukul et al. [26] tried to distinguish between ICC and HCC, or metastatic carcinoma using cytological findings of ductular clusters. They concluded that the presence of more than 10 ductular clusters associated with malignant cells was a useful discriminator to separate ICC from metastatic carcinoma but was not useful for discrimination of ICC from HCC. Therefore, the present study is the first to demonstrate that simple cytological findings can discriminate between ICC from HCC, with evidence obtained by image analysis.

## 5. Conclusions

This study evaluated the cytological characteristics of ICC and HCC using image analysis software. The results indicated that the nucleus of HCC is close to a round shape whereas the nucleus of ICC is close to an oval shape. This characteristic was significant in Papanicolaou stained cytological specimens, but this significance disappeared after Giemsa-staining. This simple and objective finding will be very useful for the differential cytodiagnosis of HCC and ICC.

## Figures and Tables

**Figure 1 cancers-14-02301-f001:**
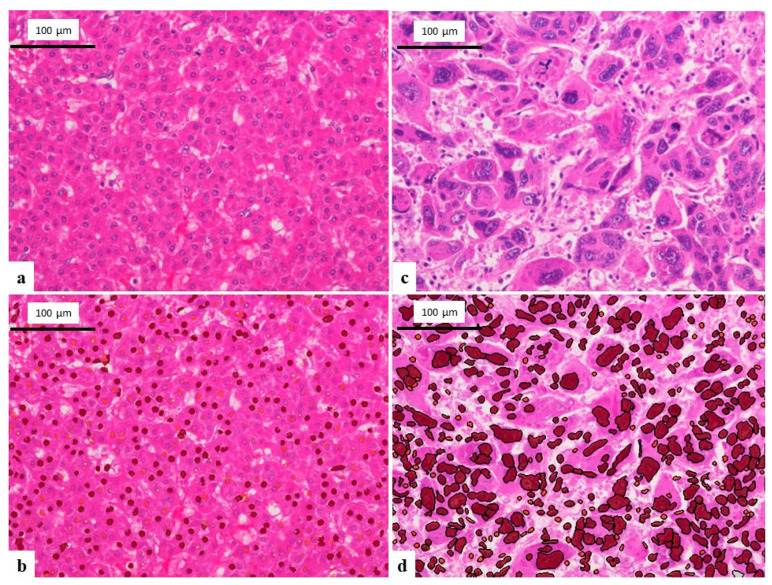
Analyzing images of HE-stained tissue section using Tissue Studio. (**a**) Image of well differentiated HCC (×200). (**b**) Analyzing image of Figure 1a. The software appropriately recognizes the nuclei of tumor cells and calculate major axis, minor axis, and area of the nucleus. (**c**) Image of poorly differentiated HCC (×200). The tumor cells are significantly larger than well differentiated HCC. (**d**) Analyzing image of Figure 1c. The software appropriately recognizes the nuclei of tumor cells although the nuclei are markedly pleomorphic.

**Figure 2 cancers-14-02301-f002:**
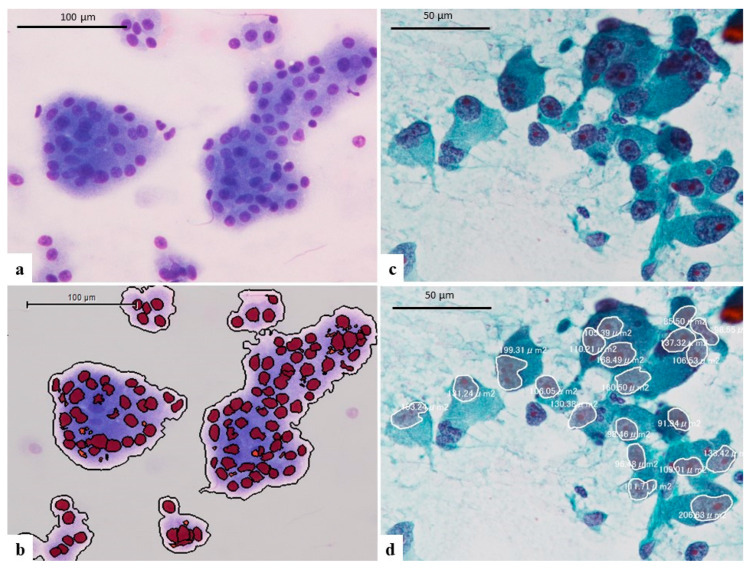
(**a**) The image of Giemsa-stained touch smear cytology of well differentiated HCC (×200, same case of Figure 1a). (**b**) Analyzing image of Figure 2a. The software Tissue Studio appropriately recognize the nuclei of tumor cells. (**c**) The image of Papanicolaou-stained touch smear cytology of poorly differentiated HCC (×200, same case of Figure 1c). (**d**) Analyzing image of Figure 2c. The nuclei were manually selected and then of the major axis, minor axis, and area of the nucleus were calculated by attaching software of EXpath III.

**Figure 3 cancers-14-02301-f003:**
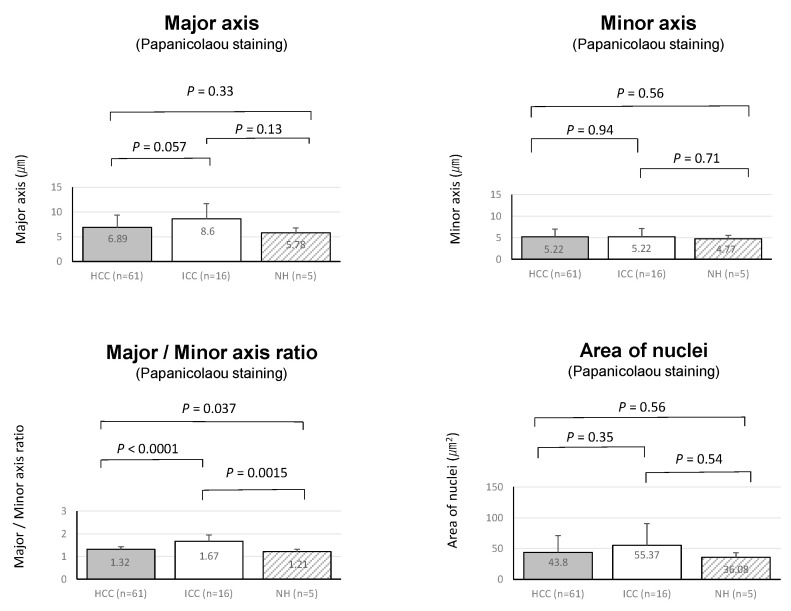
Comparison of major axis, minor axis, major/minor axis ratio and the nucleus area of hepatocellular carcinoma (HCC), intrahepatic cholangiocarcinoma (ICC) and non-tumorous hepatocytes (NH).

**Figure 4 cancers-14-02301-f004:**
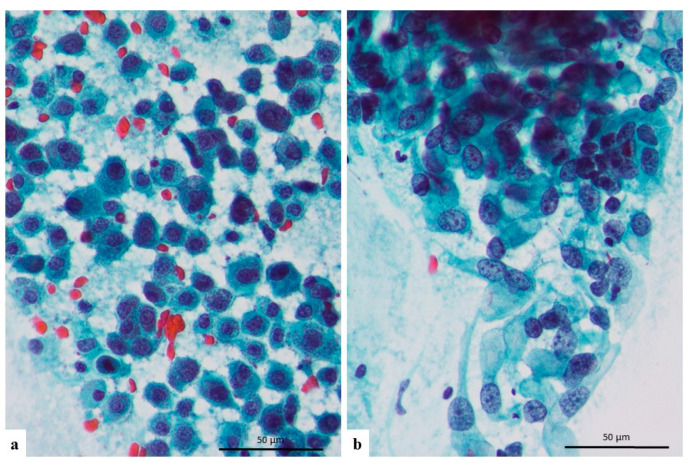
Representative cytological figures of hepatocellular carcinoma (HCC). (**a**) and intrahepatic cholangiocarcinoma (ICC). (**b**) In touch smear specimens (Papanicolaou stain, ×400). HCC cells have round and center-positioned nucleus and granular cytoplasm whereas ICC cells have oval and uncenter-positioned nucleus, multiple nucleolus, and foamy cytoplasm.

**Figure 5 cancers-14-02301-f005:**
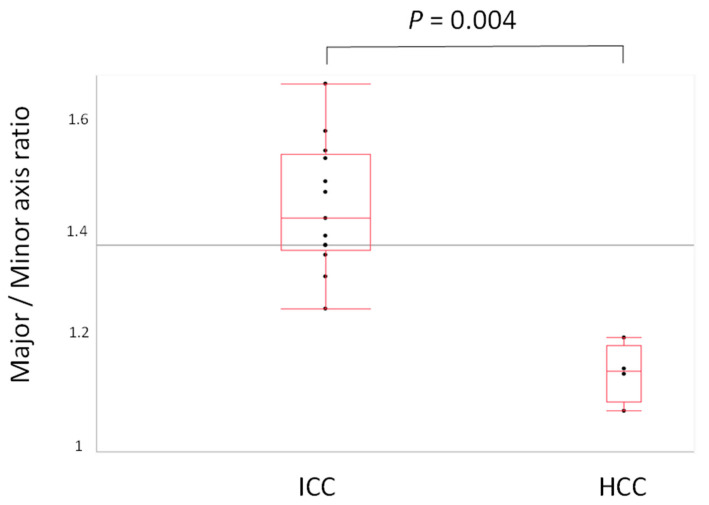
Comparison of major minor axis ratio in hepatocellular carcinoma (HCC) and intrahepatic cholangiocarcinoma (ICC) by validation Papanicolaou-stained clinical samples. A significant difference was found in the major/minor axis ratio between HCC and ICC (*p* = 0.004).

**Table 1 cancers-14-02301-t001:** Details of Touch Smear Cytology of Resected Specimens.

		Papanicolaou	Giemsa
Hepatocellular carcinoma		61	47
	Well differentiated	8	4
	Moderately differentiated	48	40
	Poorly differentiated	5	3
Intrahepatic cholangiocarcinoma		16	10
Non-tumorous liver tissue		5	0

**Table 2 cancers-14-02301-t002:** Comparison of the Nucleus between HCC and ICC for Each Stain.

	HE			Giemsa			Papanicolaou		
	HCC	ICC	*p*	HCC	ICC	*p*	HCC	ICC	*p*
Major axis (mean ± SD, µm)	11.52 ± 2.98	14.12 ± 2.05	0.0003	16.42 ± 4.47	18.46 ± 3.90	0.26	6.89 ± 2.47	8.60 ± 3.06	0.057
Minor axis (mean ± SD, µm)	8.64 ± 1.91	9.45 ± 1.23	0.031	12.04 ± 2.93	13.10 ± 2.26	0.29	5.22 ± 1.79	5.22 ± 1.95	0.940
Major/Minor axis ratio (mean ± SD)	1.36 ± 0.092	1.54 ± 0.083	<0.0001	1.39 ± 0.10	1.44 ± 0.12	0.38	1.32 ± 0.11	1.67 ± 0.27	<0.0001
Nucleus area (mean ± SD, µm^2^)	77.42 ± 38.44	93.64 ± 21.89	0.0099	146.78 ± 68.40	176.32 ± 58.20	0.14	43.80 ± 27.31	55.37 ± 35.34	0.350

HCC: hepatocellular carcinoma, ICC: intrahepatic cholangiocarcinoma, SD: standard deviation.

**Table 3 cancers-14-02301-t003:** Comparison of Cytological Fndings between ICC and HCC.

		HCC (*n* = 61)	CCC (*n* = 16)	*p*
nuclear contours (%)	irregular	23 (37.70)	2 (12.50)	0.074
	smooth	38 (62.30)	14 (87.50)	
chromatin pattern (%)	coarse/granular	34 (55.74)	6 (37.50)	0.260
	fine	27 (44.26)	10 (62.50)	
chromatin distribution (%)	homogeneous	23 (37.70)	2 (12.50)	0.074
	heterogeneous	38 (62.30)	14 (87.50)	
nuclear position (%)	center	56 (91.80)	1 (6.25)	<0.0001
	uncentre	5 (8.20)	15 (93.75)	
number of nucleolus (%)	single/unclear	52 (85.25)	8 (50.00)	0.005
	multiple	9 (14.75)	8 (50.00)	
cytoplasm (%)	vacuole/foamy	8 (13.11)	14 (87.50)	<0.0001
	granular	53 (86.89)	2 (12.50)	
cell boundaries (%)	clear	35 (57.38)	5 (31.25)	0.092
	unclear	26 (42.62)	11 (68.75)	

HCC, hepatocellular carcinoma; ICC. intrahepatic cholangiocarcinoma.

**Table 4 cancers-14-02301-t004:** Details of Clinical Specimens.

Sample Type		HCC (*n* = 4)	ICC (*n* = 13)
	FNA	4	3
	Brushing	0	8
	Ascites	0	2
**Tumor Location**			
	Primary (liver)	3	11
	Metastasis/Dissemination	1 (Lymph nodes)	2 (Ascites)

HCC: hepatocellular carcinoma, ICC: intrahepatic cholangiocarcinoma, SD: standard deviation, FNA: fine needle aspiration.

**Table 5 cancers-14-02301-t005:** Comparison of Nuclei of HCC and ICC for Each Stain.

	HCC (*n* = 4)	ICC (*n* = 13)	*p*
Major axis (mean ± SD, µm)	8.79 ± 1.58	10.01 ± 1.90	0.308
Minor axis (mean ± SD, µm)	7.67 ± 1.32	7.00 ± 1.72	0.428
Major/minor axis ratio (mean ± SD)	1.18 ± 0.056	1.45 ± 0.13	0.004
Area of nuclei (mean ± SD, µm^2^)	72.00 ± 24.87	75.49 ± 27.47	0.821

HCC: hepatocellular carcinoma, ICC: intrahepatic cholangiocarcinoma, SD: standard deviation.

## Data Availability

The data presented in this study are available on reasonable request from the corresponding author.

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
