# Peer review of "Cytological Comparison between Hepatocellular Carcinoma and Intrahepatic Cholangiocarcinoma by Image Analysis Software Using Touch Smear Samples of Surgically Resected Specimens"

_cancers, 2022, doi:10.3390/cancers14092301_

Round 1
Reviewer 1 Report
I decided to mark "accept in the present form" since after having fully evaluated the manuscript I found that:
- The topic is of interest
- English language is good and it doesn't need major revision
- Different part of the article introduction, matherials & methods, results and discussion are well exposed, concise, direct with a good resume in the tables
- Reference list ic complete.
I put accept in present form because I'm convinced of such statement and for this reason I don't see the need to add weakness and strength of the paper otherwise I'd have added revision to do on the manuscript.
Author Response
Response to reviewers
We would like to thank all the reviewers for their careful reading of our manuscript and for providing such useful feedback, which was of great help in improving the clarity of our manuscript. We have responded to all of the queries and made all requested changes to the revised manuscript. All changes based on the reviewer’s comments are highlighted.
Our point-by-point responses to the reviewer’s comments are given below.
Reviewer 1
Query/Comment 1: I decided to mark "accept in the present form" since after having fully evaluated the manuscript I found that:
- The topic is of interest
- English language is good and it doesn't need major revision
- Different part of the article introduction, materials & methods, results, and discussion are well exposed, concise, direct with a good resume in the tables
- Reference list is complete.
I put accept in present form because I'm convinced of such statement and for this reason, I don't see the need to add weakness and strength of the paper otherwise I'd have added revision to do on the manuscript.
Response: No response is needed. We would like to thank the reviewer for their time and effort in reviewing our manuscript and for the kind comments.
Reviewer 2 Report
Comment:
The manuscript solely described the difference in the nuclear morphology of HCC (round shape) and ICC (oval shape) in Papanicolaou-stained cytology specimens. The caveat was “Lines 257-262: “Therefore, we consider that the major/minor axis ratio is very useful for the differential cytodiagnosis of ICC and HCC, because of its objective nature and simpleness. The significant difference in the major/minor axis ratio of ICC and HCC samples was confirmed in HE-stained tissue specimens, but its significance disappeared in Giemsa-stained specimens, likely because of morphological changes due to the dry process involved.” “Giemsa-stained specimens” vs. “Papanicolaou-stained cytological specimens” vs. “HE staining” – Their results raised the question of whether their findings were process-specific? Such a description of morphology might not be reliable and reproducible.
Thus, even though such a morphological description is of merit and interest, this technique should be incorporated with modern technologies of clinical diagnostics for better clinical outcomes.
In conclusion: The current version is preliminary. After careful consideration, the reviewer feels that it has merit but is not suitable for publication as it currently stands. We uphold the standard and integrity of [Cancers] with an impact factor of 6.7, substantially higher than the average Journal of IF 1.0. The results of this study still need further validation research. Without the specific follow-up validation study below, this study lacks reliability and significance.
Specific comments:
- “Title: Cytological comparison between hepatocellular carcinoma and intrahepatic cholangiocarcinoma by image analysis software using touch smear samples of surgically resected specimens” This title described the methodology but lacked the thematic conclusion.
- “Touch smear specimens of surgically resected specimens were obtained from a total of 61 nodules of HCC and 16 of ICC.” The authors should clearly state how many patients each patient could have more than one nodule.
- Lines 91-101: “the imaging analysis software Tissue Studio (Definiens, München, Germany)” – any literature support for this software? What were the parameters?
- Lines 19-20: “HCC (round shape) and ICC (oval shape)” – the description is too vague: some dimensions should be helpful.
- Fig 1a &b should be marked with scale bars.
- Fig 1b. Lines 108-109, “calculate major axis, minor axis, and 108 area of the nucleus” –an illustration and a table of such calculations and statistics should be presented side-by-side with the figure.
- Figure 2. A governing title should be used before panels.
- Fig 2a &c should be marked with scale bars.
- Line 126 & Table 1 “well-differentiated HCC” – How did the authors define well differentiated? Moderately differentiated? Poorly differentiated? Any alternative cross-checking?
- Lines 132-133: “Assessment of Cytological Findings of HCC and ICC in Papanicolaou-stained Touch Smear Cytology Specimens” – repetitive of Assessment and Findings.
- Fig 2. A table of such calculations and statistics should be presented side-by-side with the figure: how many such view fields did they select?
- Fig 4 should be marked with scale bars.
- Lines 257-262: “Therefore, we consider that the major/minor axis ratio is very useful for the differential cytodiagnosis of ICC and HCC, because of its objective nature and simpleness. The significant difference in the major/minor axis ratio of ICC and HCC samples was confirmed in HE-stained tissue specimens, but its significance disappeared in Giemsa-stained specimens, likely because of morphological changes due to the dry process involved.” “Giemsa-stained specimens” vs. “Papanicolaou-stained cytological specimens” vs. “HE staining” – Their results raised the question of whether their findings were process-specific? Such a description of morphology might not be reliable and reproducible.
- Line 268: “not be safely obtained because of the patient’s general physical condition. In this situation,” Could the authors specify what such situation was?
- Lines 292-293: “The results indicated that the nucleus of HCC is close to a round shape whereas the nucleus of ICC is close to an oval shape.” – clearly, the authors did not specify how either a round shape or an oval shape formed or changed in a spatiotemporal manner of processing.
- A demographic table should be provided with patients’ outcome measurements – how did the assessment relate to clinical treatment profiles?
- How did their data relate to molecular profiling? Biomarkers? Cytogenetics? Given the single-cell clinical analysis, how could the authors incorporate it into this morphology assessment?
- Overall, the sample size was too small to conclude if this methodology has an impact clinically.
Author Response
Response to reviewers
We would like to thank all the reviewers for their careful reading of our manuscript and for providing such useful feedback, which was of great help in improving the clarity of our manuscript. We have responded to all of the queries and made all requested changes to the revised manuscript. All changes based on the reviewer’s comments are highlighted.
Our point-by-point responses to the reviewer’s comments are given below.
Reviewer 2
Query/Comment 1: “Title: Cytological comparison between hepatocellular carcinoma and intrahepatic cholangiocarcinoma by image analysis software using touch smear samples of surgically resected specimens” This title described the methodology but lacked the thematic conclusion.
Response: We have made the title for the purpose of correctly inform the methodology of our study. To the best of our knowledge, no previous study has analyzed the cytological differences between HCC and ICC by image analysis software. It is the main aim of our study.
Query/Comment 2:“Touch smear specimens of surgically resected specimens were obtained from a total of 61 nodules of HCC and 16 of ICC.” The authors should clearly state how many patients each patient could have more than one nodule.
Response: Thanks for the valuable suggestion. Six patients underwent hepatic resection for 2 HCC nodules during 1 operation. This information is now clearly documented in the revised manuscript in the Materials and Methods section (2.1.).
Query/Comment 3: Lines 91-101: “the imaging analysis software Tissue Studio (Definiens, München, Germany)” – any literature support for this software? What were the parameters?
Response: Many studies have been previously published using Tissue Studio. The parameters (µm or µm2) are documented in each Table. In the revised manuscript.
Query/Comment 4: Lines 19-20: “HCC (round shape) and ICC (oval shape)” – the description is too vague: some dimensions should be helpful.
Response: This description is supported by the results of major/minor axis ratio analysis. As the description of Lines 19-20 is a Simple Summary, we omitted the results on the major/minor axis ratio. However, this information is clearly documented in the Abstract.
Query/Comment 5: Fig 1a &b should be marked with scale bars.
Response: Thank you for the valuable suggestion. We have now added scale bars to these figures.
Query/Comment 6: Fig 1b. Lines 108-109, “calculate major axis, minor axis, and area of the nucleus” –an illustration and a table of such calculations and statistics should be presented side-by-side with the figure.
Response: The reviewer’s indication is reasonable. However, the calculated data were automatically provided by the software. Thus, we cannot not provide an illustration and a table of calculations and statistics side-by-side with the figure. Figure 1b shows that the software properly detected the nuclei of tumor cells.
Query/Comment 7: Fig 2a &c should be marked with scale bars.
Response: Thank you for your valuable suggestion. We have now added scale bars to the figures.
Query/Comment 8: Line 126 & Table 1 “well-differentiated HCC” – How did the authors define well differentiated? Moderately differentiated? Poorly differentiated? Any alternative cross-checking?
Response: The reviewer’s indication is quite reasonable. The differentiation of HCC depended on pathological reports. This description has been added to Materials and Methods section (2.1) of the revised manuscript.
Query/Comment 9: Lines 132-133: “Assessment of Cytological Findings of HCC and ICC in Papanicolaou-stained Touch Smear Cytology Specimens” – repetitive of Assessment and Findings.
Response: The assessment of Papanicolaou-stained cytology was performed by different software (EXpath III) because Tissue Studio did not support Papanicolaou staining. Therefore, we separately documented the analysis method of Papanicolaou-stained cytology other than HE and Giemsa staining.
Query/Comment 10: Fig 2. A table of such calculations and statistics should be presented side-by-side with the figure: how many such view fields did they select?
Response: The same reason for Query/Comment 6 (supra vide). Three digital images of tumor tissue (×200) were analyzed by the software. This is documented in the Materials and Methods section (2.2).
Query/Comment 11: Lines 257-262: “Therefore, we consider that the major/minor axis ratio is very useful for the differential cytodiagnosis of ICC and HCC, because of its objective nature and simpleness. The significant difference in the major/minor axis ratio of ICC and HCC samples was confirmed in HE-stained tissue specimens, but its significance disappeared in Giemsa-stained specimens, likely because of morphological changes due to the dry process involved.” “Giemsa-stained specimens” vs. “Papanicolaou-stained cytological specimens” vs. “HE staining” – Their results raised the question of whether their findings were process-specific? Such a description of morphology might not be reliable and reproducible.
Response: The cell morphology is severely affected by dry process of Giemsa-staining. Usually, the cell size is significantly enlarged, and cell morphology becomes more circular than Papanicolaou-staining. Therefore, we consider it reasonable that the results of morphological analyses by image-analyzing software were different between Giemsa-staining and Papanicolaou-staining. In addition, significance in Papanicolaou-staining is very important because clinical samples of HCC or ICC are usually evaluated by Papanicolaou-staining. This information has been added to the Discussion section of the revised manuscript.
Query/Comment 12: Line 268: “not be safely obtained because of the patient’s general physical condition. In this situation,” Could the authors specify what such situation was?
Response: We supposed a condition such as a bleeding tendency or cachexia. This information has been added to the revised manuscript.
Query/Comment 13: Lines 292-293: “The results indicated that the nucleus of HCC is close to a round shape whereas the nucleus of ICC is close to an oval shape.” – clearly, the authors did not specify how either a round shape or an oval shape formed or changed in a spatiotemporal manner of processing.
Response: The response for this comment is same as Query/Comment 11 (vide supra).
Query/Comment 14: A demographic table should be provided with patients’ outcome measurements – how did the assessment relate to clinical treatment profiles?
Response: The patient’s outcome and detail of their treatment are out with the scope of the present study. Our study focused on cytological aspects for the diagnosis of HCC and ICC.
Query/Comment 15: How did their data relate to molecular profiling? Biomarkers? Cytogenetics? Given the single-cell clinical analysis, how could the authors incorporate it into this morphology assessment?
Response: Thank you for your valuable comments. The molecular profiling, biomarkers, and cytogenetics are out of context in the present study a although correlation between cytological morphology and these factors would be of great interest. These matters will be addressed in future research.
Query/Comment 16: Overall, the sample size was too small to conclude if this methodology has an impact clinically.
Response: As described in the Introduction, fine needle aspiration cytology for liver nodule investigation is rarely performed. Thus, we have collected touch smear samples of resected specimens. Our series (Total 77 cases: 61 nodules of HCC and 16 of ICC) is a relatively large case series compared to previous reports regarding the cytology of HCC or ICC. Although the clinical sample size of cytology (such as fine needle aspiration or ascites) was small (4 cases of HCC and 13 cases of ICC), sufficient statistical significance was achieved. In addition, although the case numbers were small, numerous tumor cells were analyzed by software.
Once again, we would like to thank all the reviewers for their time and efforts in greatly improving the clarity of our manuscript.
Round 2
Reviewer 2 Report
accepted.
This manuscript is a resubmission of an earlier submission. The following is a list of the peer review reports and author responses from that submission.